# Deep Learning with a Dataset Created Using Kanno Saitama Macro, a Self-Made Automatic Foveal Avascular Zone Extraction Program

**DOI:** 10.3390/jcm12010183

**Published:** 2022-12-26

**Authors:** Junji Kanno, Takuhei Shoji, Hirokazu Ishii, Hisashi Ibuki, Yuji Yoshikawa, Takanori Sasaki, Kei Shinoda

**Affiliations:** 1Department of Ophthalmology, Saitama Medical University School of Medicine, Iruma 350-0495, Japan; 2Koedo Eye Institute, Kawagoe 350-1123, Japan

**Keywords:** foveal avascular zone, automatic extraction, manually extract, U-Net, annotation

## Abstract

The extraction of the foveal avascular zone (FAZ) from optical coherence tomography angiography (OCTA) images has been used in many studies in recent years due to its association with various ophthalmic diseases. In this study, we investigated the utility of a dataset for deep learning created using Kanno Saitama Macro (KSM), a program that automatically extracts the FAZ using swept-source OCTA. The test data included 40 eyes of 20 healthy volunteers. For training and validation, we used 257 eyes from 257 patients. The FAZ of the retinal surface image was extracted using KSM, and a dataset for FAZ extraction was created. Based on that dataset, we conducted a training test using a typical U-Net. Two examiners manually extracted the FAZ of the test data, and the results were used as gold standards to compare the Jaccard coefficients between examiners, and between each examiner and the U-Net. The Jaccard coefficient was 0.931 between examiner 1 and examiner 2, 0.951 between examiner 1 and the U-Net, and 0.933 between examiner 2 and the U-Net. The Jaccard coefficients were significantly better between examiner 1 and the U-Net than between examiner 1 and examiner 2 (*p* < 0.001). These data indicated that the dataset generated by KSM was as good as, if not better than, the agreement between examiners using the manual method. KSM may contribute to reducing the burden of annotation in deep learning.

## 1. Introduction

With the advent of optical coherence tomography angiography (OCTA), studies on the foveal avascular zone (FAZ) have been actively conducted and yielded various findings in healthy eyes [1], retinal vascular diseases (e.g., diabetic retinopathy and retinal vein occlusion) [2,3], vitreous interface lesions (e.g., epiretinal membrane and macular hole) [4,5], hereditary degenerative diseases (e.g., retinitis pigmentosa) [6], glaucoma [7] and others [8]. In these studies, the methods used to extract FAZ features included manual methods with tools for manual selection, conventional automatic methods executed by algorithms, and deep learning [9,10], which has attracted increasing attention in recent years. Although the manual method is considered the gold standard for examination because it enables more detailed extraction, it imposes a heavy burden on the examiner performing the extraction and does not guarantee reproducibility. Conventional automated methods were developed to overcome the problems associated with manual methods. These included analyses using the device’s built-in software. For example, several studies have reportedly used Python, which is a programming language and MATLAB^®^ (MathWorks) numerical analysis software, as well as ImageJ (https://imagej.nih.gov/ij, accessed on 8 February 2021), an image processing software distributed free of charge by the National Institutes of Health [11,12,13,14]. The advantage of these automated methods is that good-quality extraction can be obtained with a simple procedure. Previously, we also reported on automated extraction (Kanno Saitama Macro, KSM) using ImageJ Macro [15]. The advantage of KSM is that it can facilitate extraction that closely approximates the manual method with extremely high reproducibility with one click of a button. Furthermore, automatic extraction with a deep learning technique known as semantic segmentation is being actively promoted for medical imaging research in other specialties [16,17,18,19,20,21]. Although this method enables the simultaneous extraction of a large number of images, it requires a vast dataset and tremendous labor for the creation of the dataset (i.e., annotation). The dataset used in semantic segmentation consists of images pertaining to the question and the correct answer. In FAZ extraction, the question is the OCTA image (original image) and the correct answer is the image (label image) showing only the FAZ area. Extracting FAZ from en face images obtained with OCTA has conventionally been done manually, requiring 50 to 100 plots per image, which requires an enormous amount of time. Therefore, we investigated whether a useful data set could be created using automated methods. We used the dataset we created for training and testing using a typical U-Net. We then compared the results with the manual method to determine the usefulness of the dataset.

Although automatic extraction using artificial intelligence (AI) on healthy and diseased eyes has been introduced [9,10], to our knowledge, there are no previous reports in deep learning for FAZ extraction that aimed to automatically create FAZ datasets. Thus, we propose a method to reduce the burden of annotation using the ImageJ macro. The purpose of this study was to examine the utility of the dataset created by KSM for FAZ extraction.

## 2. Materials and Methods

### 2.1. Study Population

This study was conducted according to the Declaration of Helsinki after obtaining approval from the Saitama Medical University Hospital Ethics Committee (No. 19079.01). The study sample included 40 healthy volunteers, aged 20 years and above, who provided written informed consent for participation in the study between October and December 2017. Participants underwent comprehensive ophthalmic examinations including visual acuity measurement, visual field testing, slit-lamp examination, non-contact tonometry (TONOREFRII, Nidek, Gamagori, Japan), fundus photography (CX-1, Canon, Tokyo, Japan), axial length and central corneal thickness measurement (Optical Biometer OA-2000, Tomey Corporation, Nagoya, Japan), static visual field testing (Humphrey field analyzer, Carl Zeiss Meditec, Jena, Germany), retinal nerve fiber layer analysis using spectral-domain OCT (SD-OCT, Spectralis®HRA2, Heidelberg Engineering, Heidelberg, Germany), and swept-source OCTA (SS-OCTA) photography (PLEX^®^ Elite 9000, Carl Zeiss Meditec, Jena, Germany).

Patients with a spherical equivalent of +3 D or more or −6 D or less; axial length of 26 mm or more; suspected glaucomatous change in the visual field test, fundus photograph or retinal nerve fiber layer analysis; ocular diseases, such as diabetic retinopathy, macular disease, severe myopia, pseudoexfoliation; and those with a history of ocular surgery, were excluded. The training and validation data were obtained from each fellow eye of patients with unilateral ocular diseases (idiopathic macular hole, vitreomacular traction syndrome, glaucoma, central serous chorioretinopathy, idiopathic epiretinal membrane, and rhegmatogenous retinal detachment) who visited our clinic and underwent SS-OCTA imaging between February 2018 and September 2019. A total of 227 of 257 eyes (from 257 patients) were used to create the training dataset and the remaining 30 eyes were used to create the validation dataset. Only images with an OCTA signal strength of 8/10 or higher were incorporated into the dataset.

### 2.2. Optical Coherence Tomography Angiography

An image, measuring 3 mm × 3 mm, that was centered on the macula was acquired using SS-OCTA, with a central wavelength of 1060 nm and scanning speed of 100,000 A scan/s. Each 3 mm × 3 mm OCTA image consists of 300 pixels × 300 pixels, and is output as a 1024 pixels × 1024 pixels image. The algorithm for creating vascular signals uses optical microangiography, which measures changes in both phase and amplitude [22]. The original image used in this study was an en face image of the superficial retinal layer (SRL), defined as extending from the inner limiting membrane to the inner plexiform layer, constructed using the OCTA device’s built-in segmentation software.

### 2.3. KSM (Modified Version) and Annotation Simplification

KSM is a method that utilizes the dilation-erosion [23] morphological process, which is usually used in multiple consecutive processes, such as opening and closing, and is effective for noise reduction and edge detection [24]. In KSM, the interruptions in the vascular signal are connected with successive dilations, and the FAZ region is reproduced with successive erosions. Moreover, KSM can be customized using various processes implemented in ImageJ, since KSM is part of the ImageJ Macro. We added noise processing and changed the area expansion value to 4 pixels because the previously-reported macro did not include noise processing and had a slightly narrower extraction area. Changing these settings ensured improvements in the extraction of uneven areas and the extraction of high-brightness images (Figure 1).

The code of the setting change is presented below. (Please refer to Appendix A.)

(1)Noise processing

It was inserted in the first line of the previously-reported macro.

Run (“Bandpass Filter…”, “filter_large = 1024 filter_small = 3.5 suppress = None tolerance = 5 process”).

(2)Area expansion

The enlarged setting on the 9th line was changed to 4 pixels.

Run (“Enlarge…”, “enlarge = 4 pixels”).

Furthermore, since the previously-reported macro extracted images one-by-one, we created a macro to simplify the annotation process. In addition to the setting changes, the macro for continuous extraction was executed using the “stack” function that displays the images in the folder in one window and the “region-of-interest (ROI) set” that specifies each slice.
(1)The interpolation processing setting was changed to “none” when enlarging/reducing the image.(2)Extraction was performed with “analyze particles” instead of the wand tool and the size of the extraction area was specified.

In this study, continuous extraction was performed for every 5 images, and the ROI was saved after confirming the extraction. The procedure for dataset creation is as follows. (1) The FAZ was extracted. (2) The label image was created. (3) The label image was saved. The above-mentioned steps were repeated for the number of datasets. However, the repetition of these steps is monotonous and time-consuming even if the extraction is performed automatically. Therefore, each process was divided, and a macro of the process up to the saving step was created.

The annotation process is shown below. (Please refer to Appendix A).
(1)The folder containing the original image was loaded and displayed as a stack.(2)The FAZ was extracted from all the original images using the continuous method for every 5 images using ROI sets that specified the slices and the ROI sets were saved.(3)The entire window was selected and the “fill” command was used to suffuse all the original images with black (brightness value: 0). This image served as the background of the label image.(4)The ROI set saved in step 2 was loaded. The ROI for each slice was specified and the images were filled with white (luminance value: 255) (completion of label image).(5)The completed label images were saved one-by-one using the ROI sets specific to the slices.

The mechanism of label image creation is based on stack-based processing and extremely simple macros. Using this mechanism, dataset amplification can also be performed automatically using inversion and rotation. Creating training and validation datasets from 257 eyes, including the annotation process and FAZ extraction using KSM, required approximately 4 h—that is, approximately 1 min per eye.

Moreover, the dataset created in the above-mentioned process has a large image size of 1024 pixels × 1024 pixels, which was reduced to 512 pixels × 512 pixels to accommodate the deep learning networks. These were subsequently cropped to 256 pixels × 256 pixels.

### 2.4. Deep Learning Network

We used a typical U-Net for the semantic segmentation network [25]. The U-Net architecture is based on the fully convolutional neural network, which does not use fully-connected layers and allows images to be used as input and produces binary maps as output. As shown in Figure 2, the U-Net consists of a contracting (encoding) path and a symmetric expanding (decoding) path. In the contracting path, successive convolution layers are followed by pooling operations. In the expanding path, pooling operators are replaced by upsampling operators. The combination of the upsampled output and high-resolution features from the contracting path can supplement the information lost in the pooling process. The U-Net exhibits satisfactory performance in biomedical image segmentation because of its special structure [26]. This study used a 4-layered U-Net, binary cross entropy as the loss function, Adam [27] as the optimization algorithm, and binary accuracy as the evaluation function. Moreover, the environment was built using a graphics processing unit in Google Colaboratory Notebook. Python 3 was used as the programming language and Keras was used as the library.

### 2.5. The FAZ Extraction Method

The 3 types and 4 methods of extraction used in this study are described below.

#### 2.5.1. The Manual Method (Examiner 1 and Examiner 2)

The SRL image was imported into ImageJ. Subsequently, two examiners (H. Ibuki and H. Ishii) used the polygonal manual selection tool to trace the FAZ boundaries and save the ROI sets. An FAZ mask image was created using the above-mentioned method for the label image using the previously-obtained ROI sets.

#### 2.5.2. The Conventional Automatic Method (ARI)

The Advanced Retina Imaging Zeiss Macular Algorithm (ARI; v 0.6.1) [15] is a prototype of Carl Zeiss’s proprietary algorithm, which is available online and can be used to extract the FAZ in the SRL. Uploading an anonymized raw file to the ARI network portal causes an FAZ mask image, measuring 512 pixels × 512 pixels, to be downloaded in the Portable Network Graphics format.

#### 2.5.3. Automatic Methods Using Deep Learning (U-Net)

The dataset created by KSM was used to train and test the U-Net. First, we performed several training sessions and adjusted the number of epochs to 20 and the batch size to 12. After setting the brightness of the output image to 0 for the background and 1 for the extraction area, training and testing were performed 5 times, and all the results were acquired. The extracted image obtained was captured in ImageJ, converted into an FAZ mask image, and compared with the mask image of the manual method. The images that possessed the best results in comparison with the manual method were used in this study.

### 2.6. Evaluation of the Extraction Accuracy

The FAZ mask image obtained by each method was imported into ImageJ and converted to the same size as the extracted image obtained by the U-Net. This was followed by the evaluation of the extraction accuracy using the following indices, with the manual method as the gold standard.

#### 2.6.1. Coefficient of Variation and Correlation Coefficient of the Area

The area of the FAZ on the OCTA image was calculated using the correction formula of the magnification based on the axial length [28]. The area was quantified by inputting the measured values into a “set scale”, followed by correction. The coefficient of variation (CV) and the correlation coefficient of the area obtained, were evaluated. CV was calculated from the mean and standard deviation of the area per subject between methods.

#### 2.6.2. Measures of Similarity

The extraction accuracy is often evaluated using two measures of similarity [29,30]. However, since the evaluation differs due to the difference in the nature of the indices, both values were calculated. The similarity index evaluates the extraction target, extraction result, and the overlap between the two areas. Using the “image calculator,” we calculated and quantified the intersection and union, and the false negative (FN) and false positive (FP), and evaluated the excess and deficiency of the extraction. The above-mentioned quantification was calculated from the number of pixels in each region.

##### Jaccard Similarity Coefficient

The Jaccard similarity coefficient (Jaccard index), [11,31] which is also called Intersection over Union, is calculated by dividing the intersection of two regions (extraction target: A, extraction result: B) by the union. The results are expressed as numerical values between 1.0 to 0.0, which are graded as follows: 0.4 or less, poor; 0.7, good; and 0.9 or more, excellent.
$$\mathrm{Jaccard}\;\left(A,B\right)=\frac{A\cap B}{A\cup B}$$

##### Dice Similarity Coefficient

The Dice similarity coefficient [32,33] (DSC) is calculated by dividing the twice the value of intersection by the sum of the two regions. It is expressed as a numerical value between 1.0 to 0.0; the closer the value is to 1.0, the better the similarity. It is expressed as a higher value than the Jaccard coefficient due to the difference in the nature of the two indices.
$$\mathrm{DSC}\;\left(A,B\right)=\frac{2\left(A\cap B\right)}{A\cup B}$$

### 2.7. Statistical Analysis

The participants’ background variables were expressed as the median and interquartile range, and the FAZ area was expressed as the mean and standard deviation (SD). The CV, Jaccard coefficient, and DSC were represented as the mean and 95% confidence interval (CI). The FN and FP values were expressed as percentages (%).

We evaluated the extraction accuracy of the automatic method using the manual method as the gold standard, and also examined the accuracy between the manual methods. Nonparametric analysis was used for the obtained results since normality was rejected by the Shapiro-Wilk normality test. The area correlation coefficient was tested using Spearman’s rank correlation coefficient, and each extraction method was compared using the Friedman and multiple comparison tests (Bonferroni). The FN and FP values were compared using the Wilcoxon signed rank sum test. A *p*-value of <0.05 was considered statistically significant. All statistical analyzes were performed using the R software (version 3.6.3; R Foundation for Statistical Computing, Vienna, Austria).

## 3. Results

In this study, we used the dataset created by KSM to extract the test data (40 eyes from 20 healthy subjects) with the typical U-Net and compared the extraction results with the manual method to verify its usefulness. The participants’ background variables that were used in the test data were expressed as the median (interquartile range). The age of the target group was 30.00 (26.50 to 44.25) years. The corrected equivalent visual acuity was −0.08 (−0.08 to −0.08) logarithm of the minimum angle of resolution. The axial length was 24.15 (23.56 to 24.85) mm. The spherical equivalent was −1.25 (−2.31 to 0.00) D. Three of the 20 participants (7.5%) had a history of smoking, 1 participant (2.5%) had hypertension, and 1 participant (2.5%) had dyslipidemia; none of them had diabetes or cardiovascular disease.

### 3.1. Coefficient of Variation and Correlation Coefficient of the Area

Table 1 shows the results of the FAZ area and the Friedman test for each extraction method. Figure 3A shows the results of multiple comparisons. The area of the FAZ was 0.271 mm^2^ for examiner 1, which was significantly larger than that obtained by other methods (*p* < 0.001). The area of the FAZ for examiner 2 and the U-Net was 0.265 mm^2^, which was not significantly different (*p* = 1.00). The area of the FAZ measured using ARI has the smallest value at 0.240 mm^2^ (*p* < 0.001).

Table 2 shows the results of the CV and Friedman tests for the extraction methods and the correlation between the FAZ areas obtained with each extraction method. The results of multiple comparisons are also shown in Figure 3B. The CV was 1.61% between the manual methods (examiner 1 and examiner 2) compared to 1.35% between examiner 1 and the U-Net (*p* = 0.38), and 1.01% between examiner 2 and the U-Net (*p* < 0.001), indicating that the CV between the manual methods and the U-Net was as good as or better than that between the manual methods, with the best value for the comparison between examiner 2 and the U-Net. The results of the manual method and ARI were both higher than 4% (*p* < 0.001). The correlation coefficient showed a strong association between all the extraction methods, but the values obtained with the manual method and ARI were slightly lower.

### 3.2. Two Types of Similarity, FN and FP (Excess or Deficiency of Extraction)

Table 3 shows the similarity results and the Friedman test. Figure 4 shows the results of multiple comparisons. The Jaccard index was 0.931 between the manual methods, 0.951 between examiner 1 and the U-Net (*p* < 0.001), and 0.933 between examiner 2 and the U-Net (*p* = 1.00). The Jaccard index between examiner 1 and ARI was 0.875 (*p* < 0.001) and 0.894 between examiner 2 and ARI (*p* < 0.001). The DSC was 0.964 between the manual methods, 0.975 between examiner 1 and the U-Net, and 0.965 between examiner 2 and the U-Net. The DSC between examiner 1 and ARI was 0.933, and 0.944 between examiner 2 and ARI. The Jaccard index and DSC for the combination of the manual and the U-Net methods was equal to or higher than that for the manual methods, similar to the CV results. The best value was for the combination of examiner 1 and the U-Net, unlike the CV. Table 4 shows the results of the FN and FP quantification. FN (insufficient extraction) was significantly more common for all combinations than FP (false extraction) (*p* < 0.001), except for the combination of examiner 2 and the U-Net.

Figure 5 shows the extracted image for each method. It is apparent from the extraction results of each manual method that almost the same area was extracted by both examiners, but there was a difference in the recognition of the uneven parts (Figure 5, arrows). Figure 6 shows an image in which the extraction lines of the manual and automatic methods are superimposed. The comparison of the extraction lines of each manual method with respect to the extraction lines of the U-Net showed that examiner 1 extracted almost the same boundary as the U-Net, except for the uneven part. Conversely, the extraction of examiner 2 gave the impression of partial intersection.

## 4. Discussion

In this study, we used the dataset created by KSM to extract the test data (40 eyes from 20 healthy subjects) with the typical U-Net and compared the extraction results with the manual method to verify its usefulness. The U-Net results trained from this dataset were as good as or better than the manual results in terms of the CV of the area, correlation coefficient, and similarity evaluation. Diaz et al. [11] stated that the results of correlation coefficients between manual methods used as a gold standard will affect the performance evaluation of automatic methods. The correlation coefficient between the manual methods in this study was 0.995, which represented a strong association and seemed to be sufficiently accurate for use as the gold standard. The correlation coefficient between the manual method and ARI was also good at 0.987, but the correlation coefficient between the manual method and the U-Net was higher or equivalent to that of the than that of ARI and manual method (Table 2).

In some images, we have shown that the boundaries are different even between manual methods (Figure 5 and Figure 6). Although relatively clear images were used in this study, such errors were also observed between the manual methods. Moreover, the evaluation of the CV revealed that the combination of the manual method and the U-Net elicited the same or better results compared to the combination of the manual methods. The CVs of the manual method and ARI were more than 4%, while the CVs of the manual method and the U-Net were less than 1.5%. These findings suggest that the CVs of the manual method and the U-Net were significantly better than those of the manual method and ARI (Table 2 and Figure 3B). Similar results were obtained for the evaluation of the degree of similarity. The combination of examiner 1 and the U-Net had the best value (Table 3 and Figure 4), which differed from the results of the CV. The reason for the difference in the combination with the best values may be attributed to the nature of CV evaluation. Evaluation based on the above-mentioned characteristics of manual extraction and the results of FP and FN (Table 4) showed that the extraction of the U-Net was similar to that of examiner 1 with respect to the shape, but the area obtained with U-Net was smaller than that of examiner 1 because the FN was significantly larger than the FP in the extraction achieved by the U-Net and examiner 1 (Table 1 and Figure 3A). The area measured by the U-Net was almost the same as that of examiner 2 (Table 1 and Figure 3A), probably because there was no significant difference between the FP and FN of U-Net and examiner 2. Hence, the CV of the FAZ area was lower for the combination of the U-Net and examiner 2 than that for the combination of the U-Net and examiner 1.

Currently, reports of automated FAZ extraction include both conventional automatic methods (built-in program) [11,12,13,14,15,34] and methods using deep learning [9,10]. Table 5 presents the details of previous studies that used the Jaccard index and DSC as indicators, as well as the maximum average for each similarity [9,10,11,12,14,34]. This study was the only one to obtain an excellent (0.9 or higher) value for the Jaccard coefficient from amongst the previous studies. The lowest value was reported by Diaz et al. [11] but the correlation coefficient between the manual methods was also low in that study, which seems to be the result of the influence of the accuracy of the gold standard (as mentioned in a previous study). Moreover, ARI, which showed the lowest value in this study, also seemed to show a good result compared to previous studies.

Previous studies that employed the DSC investigated conventional automated methods and deep learning. Lin et al. [14] used Level Sets, a plugin of ImageJ, to study the extraction accuracy for images with an image quality index of 6 to 10, obtained with the Cirrus HD-OCT 5000. The extraction accuracy of Level Sets was comparable to that of the manual method, and the results were stable with various image quality levels. KSM was also used for comparison in their study. The extraction accuracy of KSM was poor at low image quality and showed inadequate reproducibility, which seemed inappropriate for the Cirrus HD-OCT 5000. The authors speculated that this was due to the false extraction caused by high-luminance noise. We assumed that the images presented in the previous study seem to be strongly affected by noise. We opine that good results can be obtained by performing noise processing (Figure 1E,F) in such cases. We recommend adjusting the number of times “dilate” and “erode” are used in the event of poor extraction, since noise processing also affects the blood flow signal. The results of Lin et al. were the lowest among the previous studies that used DSC, but even in that study, the similarity between the manual methods was also low. In other words, the accuracy of the gold standard could have affected the results in the current study, as well as that undertaken by Diaz et al. [11]. Based on the results of these two studies, there is also a need for a way to evaluate the accuracy of the gold standard in the future.

Guo et al. [9] used an improved U-Net in their study. Interestingly, that study used a dataset that included a group that edited the OCTA image and changed the brightness/contrast (B/C) to flexibly handle the extraction of OCTA images with different levels of B/C. The appeal of deep learning is that it allows for the creation of models for various conditions using datasets that have been edited to meet this purpose. Moreover, Guo et al. [9] stated that the extraction accuracy would plummet significantly in the case of conventional automatic extraction if the B/C differs from the default settings. The extraction disorder becomes stronger as the setting tolerance is exceeded in the conventional automatic method. However, images whose signal strength is reduced to the point that extraction fails are usually excluded from the study because they adversely affect the reliability of the results. Rather, the major factor that causes poor extraction seems to be a localized decrease in signal strength.

Zhang et al. [34] reported a method to deal with localized signal intensity reduction in conventional image analysis. Such local signal intensity reduction can cause extraction failure if it interferes with the FAZ. Semantic segmentation may be able to deal with local signal strength degradation that interferes with FAZ by devising the dataset. Therefore, to perform ideal extraction for various OCTA images, it is necessary to create datasets according to various requirements. To reduce the burden of creating these datasets, there is also a need for an efficient way to reduce the burden of annotation. In this study, we used ImageJ macro to simplify the annotation process; ImageJ macro is a recommended tool for annotation because it can easily automate various processes.

In the comparison of similarity, the past studies using deep learning (Guo et al. [9] and Mirshahi et al. [10]) showed good results, but this is due to the performance of the deep learning network, probably because FAZ extraction of the dataset containing the test labels was also performed by the same person. In this study, we used a typical U-Net, the FAZ extraction of the data set was performed by KSM, and the test label was extracted by the manual method. In other words, the evaluation was performed using a test label that differed from the dataset. Therefore, the results obtained in this study are excellent, and the utility of the dataset created by KSM is high.

This study has some limitations. First, all images used in the dataset, including the test data, were images with OCTA signal strength of 8/10 or higher. As shown in the study by Guo et al. [9], there are images with different luminance and B/C variations in clinical practice, and this dataset is not sufficient to deal with images with various variations. Second, the cases used for the test data included only healthy subjects. In the future, studies including diseased eyes are warranted, as in the study by Diaz et al. [11]. Regardless, the results obtained in this study are still useful based on the accuracy of the extraction and the simplification of the annotation. The next step is to evaluate the feasibility of the current method for diseased eyes. Future studies should also examine whether KSM is useful for images with lower signal strength, and whether the dataset obtained by KSM from lower signal images is useful as a dataset for deep learning. Furthermore, we aim to follow the method of Guo et al. [9] to create a dataset that can handle images with various variations. The Training: Testing ratio in this study was 8.7:1.3; Guo et al. [9] reported a ratio of 8:2, and Mirshahi et al. [10] reported a ratio of 7.7:2.3, which is close to the present study. Third, this study aimed to test the usefulness of the KSM dataset, not the performance of the neural network, and we used a typical U-Net. Other practice may have yielded different results. We plan to conduct research using other programs in the future. Fourth, another limitation is the small sample size. Further studies with a larger number of cases are needed in the future. Finally, we compared the results of the measurement method proposed in this study with those of the manual method in the same way as previous reports. The manual method is not always correct. Automated methods have reproducibility and rapidity advantages. Establishment of a measurement method that requires less manual intervention is awaited. The current study demonstrates the validity of reducing the intervention of manual methods in establishing measurement methods using AI.

## 5. Conclusions

This study demonstrated that the deep learning dataset created by KSM provides comparable performance in the extraction of FAZ with conventional automatic methods. The results can contribute to reducing the burden of annotation in deep learning and promote AI research using OCTA images.

## Figures and Tables

**Figure 1 jcm-12-00183-f001:**
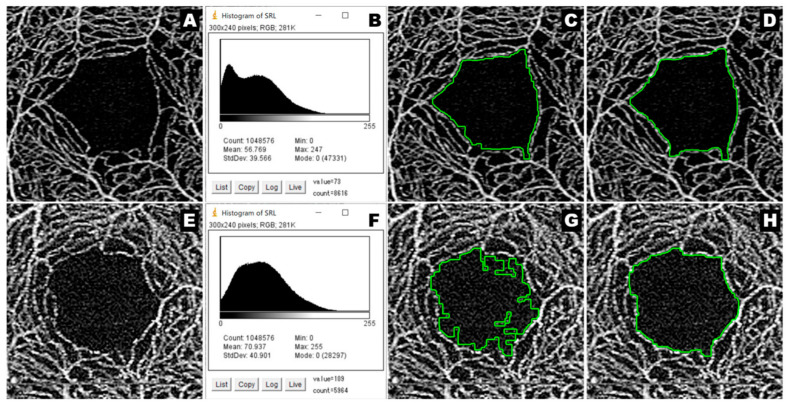
Effects of the change in KSM settings. A typical captured image (**A**) and its histogram (**B**). High-brightness images (**E**) and its histogram (**F**). Represent images extracted with the previously-reported settings (**C**,**G**), Represent images extracted after the settings were changed (**D**,**H**). PLEX^®^ Elite 9000 can acquire images with high-brightness (**E**). The histogram (**F**) of an image with high-brightness is different from the histogram (**B**) of a typical captured image (**A**). The higher the brightness of the image, the stronger the influence of noise during region extraction, thus, resulting in poorer extraction (**G**). When noise processing is added, extraction can be improved, as (**H**) is better than (**G**). Moreover, noise processing can enhance the extraction quality for typical captured images, as (**D**) is better than (**C**). KSM: Kanno Saitama Macro.

**Figure 2 jcm-12-00183-f002:**
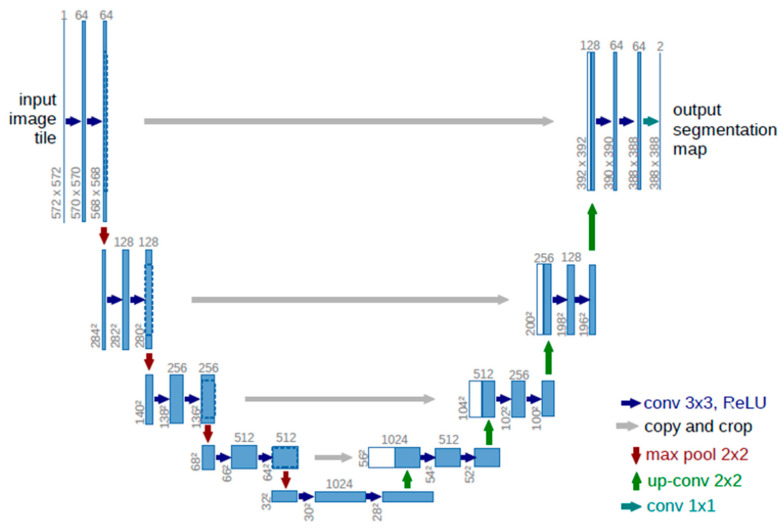
The U-Net architecture.

**Figure 3 jcm-12-00183-f003:**
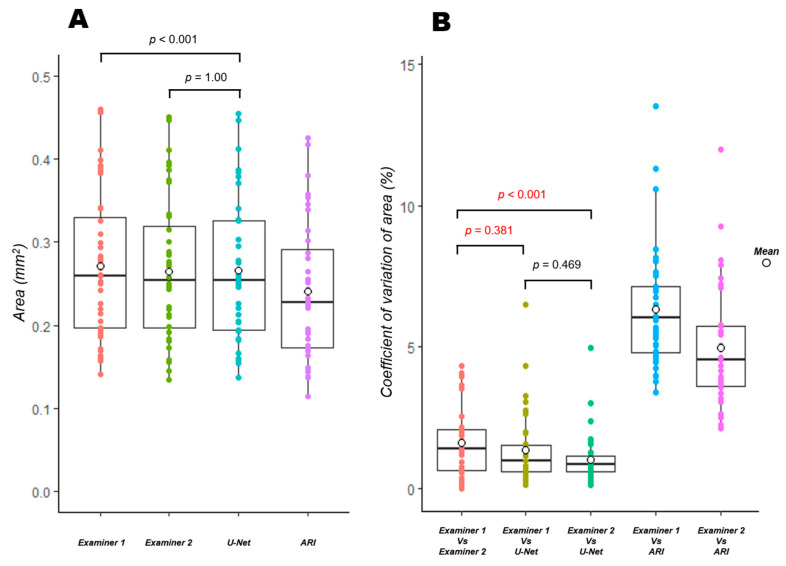
Multiple comparisons of the FAZ area and its coefficient of variation obtained by each extraction method. Results of the FAZ area (**A**) and the coefficient of variation of area (**B**) obtained by each extraction method. Comparing each combination of the four extraction methods, significant differences (*p* < 0.001) were found for all of those without a *p*-value listed. Moreover, the area obtained by U-Net is significantly different from examiner 1 (*p* < 0.001), but not from examiner 2 (**A**). (*p* = 1.00). The CV of the area between examiner 1 and the U-Net is not significantly different from that between the manual methods (*p* = 0.381), and the CV of the area between examiner 2 and the U-Net is significantly better than that between the manual methods (**B**). (*p* < 0.001). The Bonferroni correction was used to adjust the *p*-value. FAZ: foveal avascular zone.

**Figure 4 jcm-12-00183-f004:**
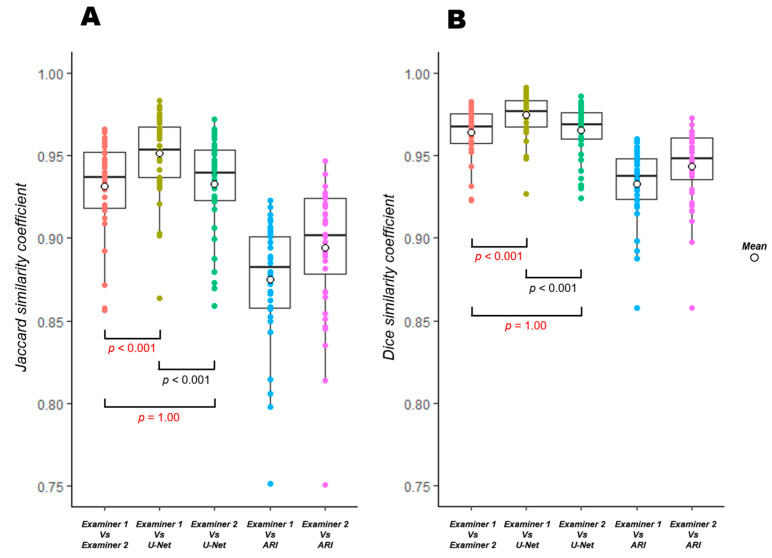
Multiple comparisons using the Jaccard index (**A**) and DSC (**B**). Results of the Jaccard index (**A**) and DSC (**B**). Comparing each 5 similarities coefficient of extraction methods, significant differences (*p* < 0.001) were found for all of those without a *p*-value listed. For similarities, the results between examiner 1 and the U-Net are significantly better than those between the manual methods (examiner 1 and 2) (*p* < 0.001), and the results between examiner 2 and the U-Net are not significantly different from those of the manual methods (*p* = 1.00). The Bonferroni correction was used to adjust the *p*-value.

**Figure 5 jcm-12-00183-f005:**
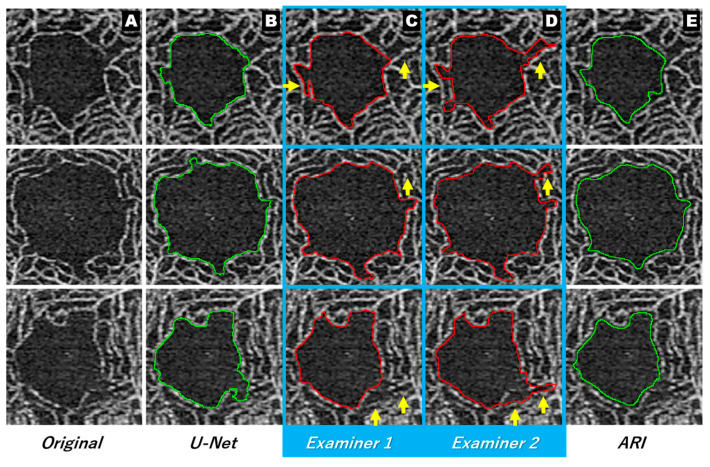
Extraction results for each method. Original image (column (**A**)), U-Net (column (**B**)). Examiners 1 and 2 (columns (**C**) and (**D**)), ARI (column (**E**)). There is a difference in the recognition of unevenness between the manual methods (Arrows in columns (**C**,**D**)). ARI: Advanced Retina Imaging.

**Figure 6 jcm-12-00183-f006:**
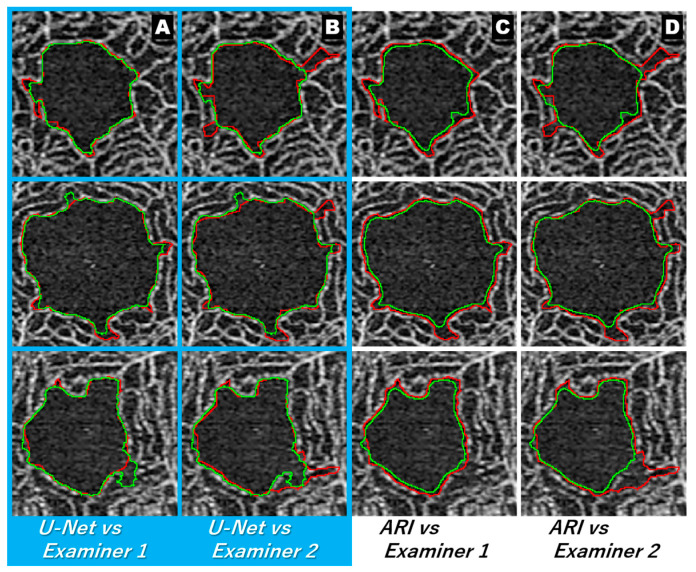
Superimposed images of the extraction lines of each automatic method and each manual method. Extraction lines for each automatic method (green line for U-Net and ARI). Extraction line for each manual method (red line). The extraction lines of examiners 1 and the U-Net demarcated almost the same area, except for the uneven area (column (**A**)). The extraction lines of examiners 2 and the U-Net appear to cross each other (column (**B**)). The extraction line for ARI looks smaller than any of the examiners (columns (**C**,**D**)). ARI: Advanced Retina Imaging.

**Table 1 jcm-12-00183-t001:** The FAZ area obtained by each extraction method and results of the Friedman test.

Method	Area (Mean ± SD) (mm^2^)
Examiner 1	0.271 ± 0.086
Examiner 2	0.265 ± 0.086
U-Net	0.265 ± 0.085
ARI	0.240 ± 0.081
*p*-value *	<0.001

* Friedman test. FAZ: foveal avascular zone, ARI: Advanced Retina Imaging.

**Table 2 jcm-12-00183-t002:** The coefficient of variation and Friedman test results for each extraction method, and correlation of the FAZ area obtained by each extraction method.

Method	CV (Mean [95%CI]) (%)	rho	*p*-Value
Examiner 1 vs Examiner 2	1.61 (1.23–1.98)	0.995	<0.001 *
Examiner 1 vs U-Net	1.35 (0.95–1.75)	0.994	<0.001 *
Examiner 2 vs U-Net	1.01 (0.73–1.29)	0.995	<0.001 *
Examiner 1 vs ARI	6.35 (5.68–7.02)	0.987	<0.001 *
Examiner 2 vs ARI	4.99 (4.33–5.65)	0.987	<0.001 *
			<0.001 ^†^

* Spearman’s rank correlation. ^†^ Friedman test. FAZ: foveal avascular zone, ARI: Advanced Retina Imaging, CV: coefficient of variation, CI: confidence interval, vs: versus.

**Table 3 jcm-12-00183-t003:** Two types of similarity and the results of each Friedman test.

	Jaccard (95%CI)	DSC (95%CI)
Examiner 1 vs Examiner 2	0.931 (0.923–0.940)	0.964 (0.959–0.969)
Examiner 1 vs U-Net	0.951 (0.943–0.959)	0.975 (0.971–0.979)
Examiner 2 vs U-Net	0.933 (0.924–0.942)	0.965 (0.960–0.970)
Examiner 1 vs ARI	0.875 (0.864–0.887)	0.933 (0.926–0.940)
Examiner 2 vs ARI	0.894 (0.881–0.906)	0.944 (0.936–0.951)
*p* value *	<0.001	<0.001

DSC: Dice similarity coefficient, ARI: Advanced Retina Imaging, CI: confidence interval, vs: versus. * Friedman test.

**Table 4 jcm-12-00183-t004:** Significant differences between the false negatives (FN) and false positives (FP).

	Mean FN (%)	Mean FP (%)	*p*-Value *
Examiner 1 vs Examiner 2	4.87	2.21	<0.001
Examiner 1 vs U-Net	3.65	1.34	<0.001
Examiner 2 vs U-Net	3.3	3.66	0.128
Examiner 1 vs ARI	12.22	0.35	<0.001
Examiner 2 vs ARI	10.07	0.68	<0.001

* Wilcoxon signed rank sum test.

**Table 5 jcm-12-00183-t005:** Previous studies using the Jaccard index and DSC, and the maximum average of each similarity.

** *Study Using the Jaccard Similarity Coefficient* **						
**Author**	**Imaging Device**	** *n* **	**Slab**	**Method**	**Maximum Mean of** **Jaccard Similarity Coefficient**	**Area Correlation Coefficient ***
Diaz et al. [11]	TOPCON DRI OCT Triton	144	SRL	Second observer	0.83	0.93
				System	0.82	0.90
Zhang et al. [34]	Optovue RTVue-XR	22	SRL	Automated Detection	0.85	
Lu et al. [12]	Optovue RTVue-XR	19	Inner Retinal	GGVF snake algorithm	0.87	
Current study	Zeiss PLEX Elite 9000	40	SRL	Second observer	0.931	0.995
				Typical U-Net (KSM Datasets)	0.951	0.994
				ARI	0.894	0.987
** *Study Using the Dice Similarity Coefficient* **						
**Author**	**Imaging Device**	** *n* **	**Slab**	**Method**	**Maximum Mean of** **Dice Similarity Coefficient**	**Area Correlation Coefficient**
Lin et al. [14]	Zeiss Cirrus HD-OCT 5000	34	SRL	Second observer	0.931	
				Level-sets macro	0.924	
				Unadjusted KSM	0.910	
Guo et al. [9]	Zeiss Cirrus HD-OCT 5000	45	SRL	Improved U-Net (Manual Datasets)	0.976	0.997
Mirshahi et al. [10]	RTVue XR 100 Avanti	10	Inner Retinal	Mask R-CNN (Manual Datasets)	0.974	0.995
Current study	Zeiss PLEX Elite 9000	40	SRL	Second observer	0.964	0.995
				Typical U-Net (KSM Datasets)	0.975	0.994
				ARI	0.944	0.987

* Correlation coefficient is the highest Index value. DSC: Dice similarity coefficient, ARI: Advanced Retina Imaging, OCT: optical coherence tomography, KSM: Kanno Saitama Macro, SRL: superficial retinal layer, R-CNN: region based convolutional neural networks.

## Data Availability

The datasets generated and/or analyzed during the current study are available from the corresponding author upon reasonable request.

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
