# Peer review of "Deep Learning with a Dataset Created Using Kanno Saitama Macro, a Self-Made Automatic Foveal Avascular Zone Extraction Program"

_jcm, 2022, doi:10.3390/jcm12010183_

Round 1

Reviewer 1 Report

1. Abstract needs revision. eg: "To investigate the utility of a dataset for deep learning created with Kanno Saitama Macro 9 (KSM), a macro for automatic extraction of the foveal avascular zone (FAZ) using swept-source optical coherence tomography angiography" - sentence is incomplete. Does not convey any meaning. 

2. Sufficient novelty is not established.

3. Unsuitable keywords. Eg: "automatic extraction of the foveal avascular zone" It may be FAZ alone

4. [1-8]. Too lengthy citation. It can be broken and paraphrased

5. Any related study work in this regard, can also be included.

6. Why U Net is used? There are advanced versions U Net in practice. ( Eg UNet++ etc)

7. What is the motivation and background behind the study?

8. On what basis the comparative study is dealt as in table 5? Do all the studies use same data splits? (testing training ratio?) If not comparison will be valid.

9. Sound conclusion expected.

10. Could not find any inference to " Optical Coherence Tomography Angiography". Clarify

11. How to validate the process/model?

12. Samples tested are too low for DL. Rigorous experimentation needed.

13. Overall Content reorganization suggested,

Author Response

Reviewer 1

  1. Abstract needs revision. eg: "To investigate the utility of a dataset for deep learning created with Kanno Saitama Macro 9 (KSM), a macro for automatic extraction of the foveal avascular zone (FAZ) using swept-source optical coherence tomography angiography" - sentence is incomplete. Does not convey any meaning. 

Response: Thank you for the suggestion. We have revised the abstract so that it can be more understandable in the following way.

“To investigate the utility of a dataset for deep learning created using Kanno Saitama Macro (KSM), which is a macro that automatically extracts foveal avascular zone (FAZ) using swept-source optical coherence tomography angiography.”

  1. Sufficient novelty is not established.

Response: We have added the following text to the introduction to indicate the novelty of this study.

“To our knowledge, there are no previous reports in deep learning for FAZ extraction that aim to automatically create FAZ datasets. Thus, we propose a method to reduce the burden of annotation using the ImageJ macro. The purpose of this study was to examine the utility of the dataset created by KSM for FAZ extraction.”

  1. Unsuitable keywords. Eg: "automatic extraction of the foveal avascular zone" It may be FAZ alone

Response: Thanks. Changed them.

  1. [1-8]. Too lengthy citation. It can be broken and paraphrased

Response: Thank you for the comments. We paraphrased the text as follows.

“With the advent of optical coherence tomography (OCTA), studies on the foveal fissure zone (FAZ) have been actively conducted and have yielded various findings include healthy eyes [1], retinal vascular diseases [2,3], vitreous interface lesions (e.g. epiretinal membrane and macular hole)[4,5], hereditary degenerative diseases (e.g. retinitis pigmentosa)[6], glaucoma [7] and other disease.[8]”

  1. Any related study work in this regard, can also be included.

Response: Thank you for the advice. The following sentence with citations has been added in the introduction.

“Although automatic extraction assisted with artificial intelligence on healthy and diseased eyes has been introduced [9,10], to our knowledge, there are no previous reports in deep learning for FAZ extraction that aim to automatically create FAZ datasets. Thus, we propose a method to reduce the burden of annotation using the ImageJ macro. The purpose of this study was to examine the utility of the dataset created by KSM for FAZ extraction.”

  1. Why U Net is used? There are advanced versions U Net in practice. ( Eg UNet++ etc)

Response: Thank you for the comments. Since this paper aims to validate the usefulness of the KSM dataset, not the performance of the neural network, a typical U-Net was used in this study. We added another limitation in the end of discussion as follows.

“Furthermore, since the purpose of this paper is to test the usefulness of the KSM dataset, not the performance of the neural network, we used a typical U-Net. Other practice may have yielded different results.”

  1. What is the motivation and background behind the study?

Response: Thank you for your thoughtful comments. In order to better clarify the motivation for this study, we have added the following text in the introduction.

“Extracting FAZ from en face images obtained with OCTA instrument has conventionally been done manually, requiring 50 to 100 plots per image, which is an enormous amount of time. Therefore, we examined whether a useful data set could be created using automated methods.”

  1. On what basis the comparative study is dealt as in table 5? Do all the studies use same data splits? (testing training ratio?) If not comparison will be valid.

Response: Thank you for the comments. We have added the following sentences in hoping it will be more understandable. 

Table 5 presents the details of previous studies that used the Jaccard index and DSC as indicators, as well as the maximum average for each similarity.

>>> 

Currently, reports of automated FAZ extraction include both conventional automatic methods [11-15,34] and methods using deep learning [9,10], and Table 5 presents the details of previous studies that used the Jaccard index and DSC as indicators, as well as the maximum average for each similarity [9-12, 14,34]. These "methods using deep learning" differ from this study in that they use manually measured data as teacher data. In this paper, automatic measurement data using macros is used as the teacher data.

  1. Sound conclusion expected.

Response: Thank you for the comments. Conclusions have been rephrased in the following way.

This study has shown that the deep learning dataset created by KSM is useful for extracting FAZ features and simplifying annotation. This result can contribute to reduce the burden of annotation in deep learning.

>>> 

This study has shown that the deep learning dataset created by KSM provides comparable performance in the extraction of FAZ with conventional automatic methods. This result may contribute to establish the AI assisted FAZ extraction method with reducing the burden of annotation in deep learning.

  1. Could not find any inference to " Optical Coherence Tomography Angiography". Clarify

Response:  The extraction of FAZs, which is the main focus of this study, is based on the analysis of images obtained with OCTA. To clarify the role of OCTA in this study, we have added the following text to the introduction.

“Extracting FAZ from en face images obtained with OCTA instrument has conventionally been done manually, requiring 50 to 100 plots per image, which is an enormous amount of time. Therefore, we examined whether a useful data set could be created using automated methods.”

  1. How to validate the process/model?

Response : Thank you for the comments. In this study, we compared the results of the measurement method proposed in this study with those of the manual method, similar to the previous report presented in Table 5. Although the results of the manual method are not always correct, We believe that the above comparison is valid because the manual method is the realistic gold standard for practical use.

According to the rewiewer's comments, we added the following sentence as a limitation.

“In this study, we compared the results of the measurement method proposed in this study with those of the manual method in the same way as previous reports. But the manual method is not always perfectly correct. Automated methods have advantages in terms of reproducibility and rapidity. Establishment of a measurement method that requires less manual intervention is awaited. We believe that the current study demonstrates the validity of reducing the intervention of manual methods in establishing measurement methods using AI”

  1. Samples tested are too low for DL. Rigorous experimentation needed.

Response:  In accordance with the reviewers' concerns, the small sample size was added as a limitation to this study.

“We are aware that another limitation is the small sample size. Further studies with a larger number of cases may be needed in the future.”

  1. Overall Content reorganization suggested,

Response: Thank you for comments. We have reorganized the content based on the reviewer's thoughtful comments as noted above.

Reviewer 2 Report

Please clearly state in the article whether the clinical experience of the two examiners who manually labeled the FAZ area is the same, thank you!

Author Response

Reviewer 2

Please clearly state in the article whether the clinical experience of the two examiners who manually labeled the FAZ area is the same, thank you!

Response: Thank you for the comments. As per reviewer’s suggestion, we changed term to “two retina expert examiners”

Reviewer 3 Report

Comments to the author:

Thank you very much for this engaging and well-written paper 

As the author acknowledges, this study's main limitation is the use of all images with a signal strength of 8/10 or higher, and the use of just healthy patients. It could be interesting to understand what happens with lower signal images. 

It could be useful to analyze the mean results of the two manual examiners and the DSC and ARI, if possible. 

P1 lines 26: I prefer foveal avascular zone 

Author Response

Reviewer 3

Thank you very much for this engaging and well-written paper 

As the author acknowledges, this study's main limitation is the use of all images with a signal strength of 8/10 or higher, and the use of just healthy patients. It could be interesting to understand what happens with lower signal images. 

 Response: Thank you for the comments. We understand the point.

As the reviewer indicates, it was not difficult to collect data with signal strength more than 8/10 from healthy subjects. But the goal is to establish an AI assisted method applicable to diseased eyes.

We are thinking of several ways to develop the AI assisted method. Besides following the method of Guo et al, one possible way would be to examine the feasibility of the current method for diseased eyes. Another way worth trying would be to examine whether KSM is useful on the images with lower signal strength, and then, to examine whether the dataset obtained by KSM from lower signal images is useful as a dataset for deep learning. This has been added to the limitation paragraph in the discussion section as follows.

First of all, it means that all images used in the dataset, including the test data, were images with OCTA signal strength of 8/10 or higher. As shown in the study by Guo et al, [9] there are images with different luminance and B/C variations in clinical practice, and this dataset is not sufficient to deal with images with various variations. In addition, the cases used for the test data were only healthy subjects, so it will be necessary to include diseased eyes in the future, as in the study by Diaz et al. [11] However, the results obtained in this study seem to be very useful for the accuracy of the extraction and the simplification of the annotation. It would be worth trying next to find out the feasibility of the current method for diseased eyes. Another way would be to examine whether KSM is useful on the images with lower signal strength, and then, to examine whether the dataset obtained by KSM from lower signal images is useful as a dataset for deep learning. Furthermore, we will follow the method of Guo et al. [9] to create a dataset that can handle images with various variations.

It could be useful to analyze the mean results of the two manual examiners and the DSC and ARI, if possible. 

 Response: Thank you for the comments. If we understand the reviewer's suggestion correctly, we think unfortunately it is not possible. Since similarity (DSC and Jaccard) shows the difference between the two shapes, it is not possible to calculate the mean results of the two manual examiners.

P1 lines 26: I prefer foveal avascular zone 

Response:  Thank you. We changed it.

Round 2

Reviewer 1 Report

still there are some areas of concern. Firstly proof reading is strongly recommended. First statement of abstract starts with "To". Its like writing an objective or aim. It must be like " .....is investigated"

Why U-Net is preferred? Other improved version of U-Net have shown promising results. Any comparison would be useful.

On comparing previous works, whether all those compared ones, use the same dataset? Same set of data splits? ( Training : Testing ratio)

Conclusion can be much strong.

Author Response

Reviewer 1

  1. still there are some areas of concern. Firstly proof reading is strongly recommended. First statement of abstract starts with "To". Its like writing an objective or aim. It must be like " .....is investigated"

Response: Thank you for the suggestion. As per the reviewer’s suggestion, we changed them.

“To investigate the utility of a dataset for deep learning created using Kanno Saitama Macro (KSM), which is a macro that automatically extracts foveal avascular zone (FAZ) using swept-source optical coherence tomography angiography”

>> 

“Extraction of foveal avascular zone (FAZ) from optical coherence tomography (OCTA) images has been used in many studies in recent years due to its association with various ophthalmic diseases. In this study, we investigated the utility of a dataset for deep learning created using Kanno Saitama Macro (KSM), which is a macro that automatically extracts foveal avascular zone (FAZ) using swept-source OCTA.”

We have again edited and checked the grammar and added the following to the Acknowledgements.

Acknowledgments: We would like to thank Editage (www.editage.com) for English language editing.

  1. Why U-Net is preferred? Other improved version of U-Net have shown promising results. Any comparison would be useful.

Response: Thank you for the suggestion. As we replied in the first round, the purpose of this study was not to compare the performance of deep learning programs, but to verify whether a dataset that was automatically FAZ extracted is feasible as teacher data. We will plan to conduct research using other programs in the future.

The following text was added to the limitations section of the discussion.

“Third, this study aimed to test the usefulness of the KSM dataset, not the performance of the neural network, and we used a typical U-Net. Other practice may have yielded different results. We will plan to conduct research using other programs in the future.”

  1. On comparing previous works, whether all those compared ones, use the same dataset? Same set of data splits? ( Training : Testing ratio)

Response: Thank you for the suggestion. Based on the reviewer's suggestion, we compared our study with previous reports. The Trainig: Testing ratio in this study was 8.7:1.3; Guo et al. reported a ratio of 8:2, and Mirshahi et al. reported a ratio of 7.7:2.3, which is close to the present study.

The following text was added to the limitations section of the discussion.

>> 

“Furthermore, we aim to follow the method of Guo et al. [9] to create a dataset that can handle images with various variations. The Trainig: Testing ratio in this study was 8.7:1.3; Guo et al. [9] reported a ratio of 8:2, and Mirshahi et al. [10] reported a ratio of 7.7:2.3, which is close to the present study.”

  1. Conclusion can be much strong.  

Response: Thank you for the suggestion. We emphasized our conclusions as follows.

This result can contribute to establish the AI assisted FAZ extraction method with reducing the burden of annotation in deep learning.

>> 

The results can contribute to reducing the burden of annotation in deep learning and promote AI research using OCTA images.